# An optimized initialization for LDPC decoding over GF(q) in impulsive noise environments

**Haoqiang Liu[1], Hongbo Zhao[1,2]\*, Xiaowen Chen[1], Wenquan Feng[1]**

**1** School of Electronic and Information Engineering, Beihang University, Beijing, P. R. China, **2** Hefei Innovation Research Institute of Beihang University, Beihang University, Beijing, P. R. China

\* bhzhb@buaa.edu.cn

## Abstract

Modern navigation satellite communication has the characteristic of high transmitting rate. To avoid bit errors in data transmission, low density parity check (LDPC) codes are widely recognized as efficient ways for navigation communication. Conventionally, the LDPC decoding is applied for additive white Gaussian noise (AWGN) channel and degrades severely while facing the impulsive noise. However, navigation communication often suffers from impulsive interference due to the occurrence of high amplitude "spikes". At this time, the conventional Gaussian noise assumption is inadequate. The impulsive component of interference has been found to be significant which influences the reliability of transmitted information. Therefore the LDPC decoding algorithms for AWGN channel are not suitable for impulsive noise environments. Consider that LDPC codes over GF($q$) perform better than binary LDPC in resisting burst errors for current navigation system, it is necessary to conduct research on LDPC codes over GF($q$). In this paper, an optimized initialization by calculating posterior probabilities of received symbols is proposed for non-binary LDPC decoding on additive white Class A noise (AWAN) channel. To verify the performance of the proposed initialization, extensive experiments are performed in terms of convergence, validity, and robustness. Preliminary results demonstrate that the decoding algorithm with the optimized initialization for non-binary LDPC codes performs better than the competing methods and that of binary LDPC codes on AWAN channel.

## Introduction

With the explosive development of communication technology, the new mobile communication systems, such as beyond fifth generation (B5G) and sixth generation (6G) systems, will suffer from severe challenges imposed by the requirement for heavy connection density and high efficiency [1]. Especially for 6G, satellite communications play an important role in providing high quality communication services to achieve the worldwide connectivity [2]. As one of the critical components, modern navigation satellite communication has the characteristic of high transmitting rate which is a real challenge to ensure the correctness of transmitted information in various channels [3,4]. To avoid bit errors in data transmission, it is an efficient way to employ low density parity check (LDPC) codes in navigation satellite communication

**Data Availability Statement:** All relevant data are within the manuscript and its Supporting Information files.

**Funding:** This paper was supported by the National Natural Science Foundation of China (No.91438116 and No.61901015). The funders had no role in

study design, data collection and analysis, decision to publish, or preparation of the manuscript.

channel. Nowadays, LDPC codes have been widely applied in the 2nd-generation digital video broadcast via satellite (DVB-S2) standard [5], 5G mobile communications [6], DNA barcoding [7], as well as the uplink of mobile satellite communication [8].

For modern navigation system, LDPC codes over $GF(q)$ have been employed in encoding and decoding as important codes. With the rapid development of BeiDou Navigation Satellite System (BDS) 3, B1C signal and B2a signal have been utilized gradually. The B-CNAV1 navigation message is transmitted through B1C signal. Each frame of message consists of 3 sub frames, of which the second sub frame exploits LDPC codes (200,100) over GF(64) and the third sub frame leverages LDPC codes (88, 44) over GF(64) for encoding [9]. Meanwhile, the B-CNAV2 navigation message is transmitted through B2a signal with LDPC codes (96, 48) over GF(64). Non-binary LDPC codes have been recognized as a powerful technology to encode navigation information efficiently. With the continuous development of BDS 3 and the increasing demand for navigation, people hope to access precise navigation information in time. However, working in severe environments, such as underwater environment surrounded by acoustical noises or in the power industry, the decoder of non-binary LDPC codes is inevitably subject to impulsive interference, which causes poor reliability of transmitted information and even leads to communication failure. Therefore, nowadays, evaluation and analysis of LDPC codes over $GF(q)$ for navigation satellite communication, especially in harsh environments, arouse attentions of researchers all over the world.

Conventionally, the LDPC decoding algorithms are designed for AWGN channel. However, navigation satellite communication often suffers from irregular noises due to the occurrence of high amplitude "spikes". For navigation satellite communication, such spikes can be generated in atmosphere where lightning discharges in the vicinity of the receiver, or underwater environment where the ambient acoustical noises includes impulses due to noisy aquatic animals such as snapping shrimp [10–13]. At this time, the original assumption that noises are the Gaussian noises is inadequate. Unfortunately, few attention has been given to decoding algorithms for LDPC codes over $GF(q)$ in impulsive noise environments.

Generally, compared with binary LDPC, LDPC codes over $GF(q)$ show superior performance in resisting burst errors such as impulsive noises for the characteristic of inner interleaving [14,15], which makes non-binary LDPC codes more suitable for navigation communication. Furthermore, non-binary LDPC codes combined with q-ary modulation can increase transmission rate obviously [16,17]. Although there is an increment of computational complexity by adopting LDPC codes over $GF(q)$, with the development of terminal computation, non-binary LDPC codes have become a hot research topic and would become more prevalent and applicable. Therefore, we focus on non-binary LDPC codes decoding on additive white Class A noise (AWAN) channel and present an optimized initialization for decoding in this paper. With simulation experiments, we demonstrate the efficiency of the proposed algorithm. The main contributions are summarized as follows:

1. We investigate the problem of LDPC decoding in impulsive noise environments for navigation communication and formalize the impulsive noise as the Class A noise model.

2. We propose an optimized initialization by calculating posterior probabilities of received symbols for non-binary LDPC decoding on AWAN channel, which makes use of series truncation for computing effectively.

3. Extensive experiments are conducted on convergence, validity, and robustness. The experimental results reveal that the optimized initialization has a significant effect on the decoding performance for non-binary LDPC codes on AWAN channel.

The rest of the paper is organized as follows. We illustrate the related work in Section 2. Section 3 describes AWAN model based on the statistics and conventional initialization of decoder for LDPC codes over GF($q$). Section 4 illustrates the optimized initialization process on AWAN channel. Simulation results are shown and discussed in Section 5. Finally, the conclusion is given in Section 6.

## Related work

LDPC codes were introduced by Gallager in 1962 for the first time, which are linear codes with sparse parity-check matrix [18]. In 1996, Mackay and Wiberg found that LDPC codes achieved excellent performance approximating the Shannon limit, and it soon became a research hotspot in the channel coding theory [19]. To improve the performance of error correction and transmission, Davey and MacKay extended the belief-propagation (BP) decoding algorithm for binary LDPC to non-binary LDPC firstly [20]. Furthermore, they proposed a fast Fourier Transform-based belief-propagation (FFT BP) decoding algorithm for reducing complexity [21]. Unfortunately, this method was only valid when the Galois field was a binary extension field with $q = 2^p$ and inefficient to handle other situations. To further simplify the decoding procedure, Declercq et al. introduced an extended min-sum algorithm, but it led to degradation of performance [14]. The United States adopted LDPC codes with 1/2 code rate in the L1C signal of GPS for the first time. Hareedy Ahmed et al. proposed non-binary LDPC codes for magnetic recording channels, and provided a comprehensive analysis of the error floor along with codes optimization guidelines for structured and regular non-binary LDPC codes [22]. In 2017, Huang Qin proposed message-passing decoding algorithms that decoded non-binary LDPC codes including ultra-sparse ones efficiently [23]. In 2019, Rehman employed parallel architecture for LDPC codes decoding to achieve the higher data rate, which in turn raised the memory conflict issue [24]. In recent years, with the emergence of decoding algorithms with low complexity, LDPC codes show superiority in practice with their excellent capability, and have been replacing the conventional codes as main codes in future navigation satellite communication gradually.

Generally, receivers adopts parameters of the AWGN channel and conventional decoding methods for LDPC decoding, which results in serious degradation in impulsive noise environments. The impulsive component of interference has been found to be significant which influences the reliability of transmitted information.

Various attempts have been made to develop models of impulsive noises that can be divided into empirical models and physical models. Class A noise model proposed by Middleton is a typical kind of physical models [25]. The statistical feature of Class A noise is much different from that of Gaussian noise, therefore the LDPC decoding algorithms for AWGN channel are not suitable for Class A noise environments. Maad et al. analyzed the performance of LDPC codes in heavy-tailed, symmetric alpha stable noise (SαS) channels [26]. Further, Nakagawa et al. proposed the sum-product decoding method for binary LDPC codes in Class A noise environment [27]. However, few researches have ever explored decoding algorithms for LDPC codes over GF($q$) in Class A noise environments.

This paper is focused on the design of an optimized initialization for non-binary LDPC codes on AWAN channel. In contrast to metaheuristics, our optimized initialization is dedicated to deal with decoding problems based on BP. Generally, as the famous optimization techniques, metaheuristics are widely recognized as efficient approaches for optimization problems, such as particle swarm optimization (PSO) [28,29] and differential evolution (DE) [30]. Various metaheuristic methods have reported advantages in image segmentation [31], tuning hyper-parameters of deep neural networks [32,33], and benzene prediction model [34]. However, as illustrated in [35], the successful application of metaheuristics requires to find a

good initial parameter setting, which is a tedious and time consuming task. Moreover, the performance of metaheuristics deteriorates quickly as the dimensionality increases, nevertheless high-dimensional circumstances are extremely common in encoding and decoding. Therefore, decoding algorithms based on BP are considered for non-binary LDPC codes on AWAN channel, and an optimized initialization by calculating posterior probabilities of received symbols is proposed.

## Theoretical background

### AWAN model

When encountering lightning or water during transmission, navigation signals will be interfered by the impulse noise, which leads to the unexpected change of the amplitude of the transmitted signal. In this part, the Class A noise model devised by Middleton [25] is described as a statistical AWAN model with the impulsive noise environment, which is widely applicable by adjusting parameters and provides fine closeness to experimental values. According to this theory, the Class A noise model is composed of Gaussian noise $G(t)$ and impulsive noise $X(t)$, which can be expressed as

$$N(t) = X(t) + G(t), \tag{1}$$

where $G(t)$ is considered as background noise and $X(t)$ can be expressed as [25]

$$X(t) = \sum_j U_j(t, \vartheta). \tag{2}$$

Here, $U_j$ denotes the $j$-th received impulse noise waveform from an interfering source and $\vartheta$ represents the random parameters which describe the waveform scale and structure. Assume that there is only one type of waveform, and $U$ is generated with appropriate variations in the individual wave form under the variation of the parameter $\vartheta$. According to the Class A noise model, the probability density function (PDF) of the noise amplitude $z$ can be defined as

$$P_A(z) = \sum_{m=0}^{\infty} \frac{e^{-A} A^m}{m!} \cdot \frac{1}{\sqrt{2\pi}\sigma_m} \exp\left(-\frac{z^2}{2\sigma_m^2}\right), \tag{3}$$

where $\sigma_m^2 = \sigma^2 \cdot (m/A + \Gamma)/(1 + \Gamma)$, $A$ is the impulsive coefficient which is defined as average number of impulses on the receiver in unit time. $\Gamma = \sigma_G^2/\sigma_I^2$ is the Gaussian-to-Impulsive noise power ratio (GIR) with Gaussian noise power $\sigma_G^2$ and impulsive noise power $\sigma_I^2$. Therefore the total noise power is $\sigma^2 = \sigma_G^2 + \sigma_I^2$.

The Class A noise in (3) consists of the impulsive noise with variance $\sigma_I^2$ and the background Gaussian noise with variance $\sigma_G^2$. The number of impulsive noise is distributed with Poisson distribution $(e^{-A} \cdot A^m)/m!$ and the amplitude of each impulsive noise is characterized by a Gaussian PDF with variance $\sigma_I^2/A$. Therefore, at a certain observation time, assume that the number of impulsive noise is $m$, which is characterized by a Poisson distribution with mean $A$, the noise of receiver is characterized by a Gaussian PDF with variance $\sigma_m^2 = \sigma_G^2 + \sigma_I^2 \cdot m/A$.

Consider the independence between the background Gaussian noise and the impulsive noise, as the impulse coefficient $A$ increases, the impulsive noise becomes more intensive and continuous, which makes the Class A noise approximate the Gaussian noise. In particular, if $A$ is close to 10, the statistical feature of the Class A noise is almost similar to that of the Gaussian noise [25]. In addition, as $\Gamma$ grows, i.e., the proportion of Gaussian white noise in the total noise increases, the Class A noise gets close to the Gaussian noise. Otherwise, the smaller $\Gamma$ is, the more impulsive the Class A noise would be.

## Conventional initialization in decoding process

Non-binary LDPC codes can be considered as a kind of linear block codes which are the extension of binary LDPC codes over GF($q$). The difference between non-binary LDPC codes and binary LDPC codes is that each non-zero element of sparse parity-check matrix needs to be obtained from GF($q$). The Tanner graph of non-binary LDPC codes given by a sparse parity-check matrix over GF($q$) is constructed in the same way as that of binary LDPC codes. Compared with binary LDPC codes, non-binary LDPC codes perform better in communication due to advantages in resisting burst error and high transmission rate [36–39].

A significant amount of research has been concentrated on the design, encoding, decoding and performance analysis of non-binary LDPC codes. Iterative decoding algorithm based on BP is an important soft decision decoding algorithm. According to this algorithm, messages are delivered between variable nodes and check nodes during iterations after initialization. Further, the received codes are updated until satisfying the parity-check equations or the upper limit of the iteration number is reached.

Since the channel transition probability is only utilized in the initialization process, we focus on optimizing initialization of LDPC decoding on AWAN channel. In this subsection, traditional initialization method for non-binary decoding is introduced. Consider that the codeword $u = (u_0, u_1, \ldots, u_{n-1})$ is obtained by encoding information sequence $e = (e_0, e_1, \ldots, e_{k-1})$. $Q$-ary sequence $u$ can be expanded into binary sequence $((u_{0,1}, u_{0,2}, \ldots, u_{0,p}), \ldots, (u_{n-1,1}, u_{n-1,2}, \ldots, u_{n-1,p}))$ and modulated by binary phase shift keying (BPSK) (using the mapping 0 to 1, 1 to -1) to obtain the transmitted sequence $x = ((x_{0,1}, x_{0,2}, \ldots, x_{0,p}), \ldots, (x_{n-1,1}, x_{n-1,2}, \ldots, x_{n-1,p}))$, where $p = \log_2 q$. And $y = ((y_{0,1}, y_{0,2}, \ldots, y_{0,p}), \ldots, (y_{n-1,1}, y_{n-1,2}, \ldots, y_{n-1,p}))$ is the received sequence transmitted on AWGN channel with variance $\sigma^2$ and mean 0.

The initial messages sent from variable node $v_j$ to the check node $c_i$ is the probabilities of the $j$-th code symbol $u_j$ equal to $\{a_1, a_2, \ldots, a_q\}$ respectively, given received sequence $y$. And it can be expressed as

$$P_j^a = P(u_j = a_k | y), \tag{4}$$

where $1 \leq k \leq q$ and $a_k$ denotes the element of GF($q$). We expand $a_k$ to binary sequence $\{a_{k1}, a_{k2}, \ldots, a_{kp}\}$ and the Eq (4) can be rewritten as:

$$
\begin{aligned}
P_j^a &= P(u_{j,1} = a_{k1}, \cdots, u_{j,p} = a_{kp} | y_{j,1}, \cdots, y_{j,p}) \\
&= P(u_{j,1} = a_{k1} | y_{j,1}) \cdot P(u_{j,2} = a_{k2} | y_{j,2}) \cdots P(u_{j,p} = a_{kp} | y_{j,p})
\end{aligned} \tag{5}
$$

There is a mapping between expanded codeword $u$ and transmitted sequence $x$ due to BPSK modulation. Thus, $P(u_{j,1} = a_{k1} | y_{j,1})$ equals to $P(x_{j,1} = 1 - 2a_{k1} | y_{j,1})$. According to the Bayes formula, the posterior probabilities of received bits can be acquired by

$$
\begin{aligned}
P(x_{j,l} = a | y_{j,l}) &= \frac{P(x_{j,1} = a) \cdot P(y_{j,1} | x_{j,1} = a)}{P(x_{j,l} = a) \cdot P(y_{j,1} | x_{j,1} = a) + P(x_{j,1} = -a) \cdot P(y_{j,1} | x_{j,1} = -a)} \\
&= \frac{1}{1 + \dfrac{P(y_{j,1} | x_{j,1} = -a)}{P(y_{j,1} | x_{j,1} = a)}} \\
&= \frac{1}{1 + \dfrac{P_G(y_{j,1} + a)}{P_G(y_{j,1} - a)}}
\end{aligned} , \tag{6}
$$

where $a = \pm 1$, $P(x_{j,1} = \pm a) = 0.5$ and $P_G(\cdot)$ is the Gaussian PDF with mean 0 and variance $\sigma_G^2$.

Due to

$$\lambda_G(y_{j,1}) = \ln\frac{P_G(y_{j,1} + a)}{P_G(y_{j,1} - a)} = -a \cdot \frac{2y_{j,1}}{\sigma_G^2},\tag{7}$$

thus

$$P(x_{j,1} = a|y_{j,1}) = \frac{1}{1 + \exp(-a \cdot \lambda_G(y_{j,1}))}.\tag{8}$$

And the initial messages of BP algorithm for LDPC codes over GF($q$) could be obtained by adding (8) to (5).

## An optimized initialization for LDPC decoding over GF($q$) on AWAN channel

The conventional decoding methods degrade severely while facing the impulsive noise since those methods acquire the posterior probability of received symbols by making use of the transition probability on AWGN channel. To tackle this problem, based on the BP decoding algorithm, we present an optimized initialization for LDPC decoding over GF($q$) on AWAN channel with series truncation.

As mentioned above, the initialization is the crucial part of the BP decoding over GF($q$) algorithm. The posterior probability of received symbols is considered as the initial message transmitted from the variable nodes to the check nodes. Thus, to evaluate the posterior probability of received symbols correctly is vital for decoding.

During the initialization of conventional decoding process, we obtain the posterior probability of the received bits from the transition probability of the AWGN channel as Eq (9), which suffers from degradation of the bit error rate (BER) in impulsive noise environments. Therefore, for LDPC decoding over GF($q$) on AWAN channel, since the channel parameters are only leveraged in the decoding initialization, we need to improve the initialization of the iterative decoding process without changing the information transmission mode between variable nodes and check nodes. Under this circumstance, the optimized initialization can be applied to various traditional LDPC decoding algorithms over GF($q$), such as FFT BP and EMS algorithm. Generally, the BP decoding algorithm with optimized initialization for LDPC codes over GF($q$) on AWAN channel can be illustrated by the flow chart in Fig 1.

In the following part, we elaborate the optimized initialization in FFT BP decoding algorithm and series truncation is proposed to calculate the PDF of the Middleton Class A noise effectively. The LLR of the received bit can be defined as

$$\lambda_A(y_{j,k}) = \ln\frac{P(x_{j,k} = +1|y_{j,k})}{P(x_{j,k} = -1|y_{j,k})},\tag{9}$$

where $0 \leq j < n$ and $1 \leq k \leq p$. The posterior probability in Eq (9) can be expressed as prior probability by

$$\begin{aligned}
\log\frac{P(x_{j,k} = +1|y_{j,k})}{P(x_{j,k} = -1|y_{j,k})} &= \log\frac{P(x_{j,k} = +1, y_{j,k})}{P(x_{j,k} = -1, y_{j,k})}\\
&= \log\frac{P(y_{j,k}|x_{j,k} = +1)P(x_{j,k} = +1)}{P(y_{j,k}|x_{j,k} = -1)P(x_{j,k} = -1)},\\
&= \log\frac{P(y_{j,k}|x_{j,k} = +1)}{P(y_{j,k}|x_{j,k} = -1)}
\end{aligned}\tag{10}$$

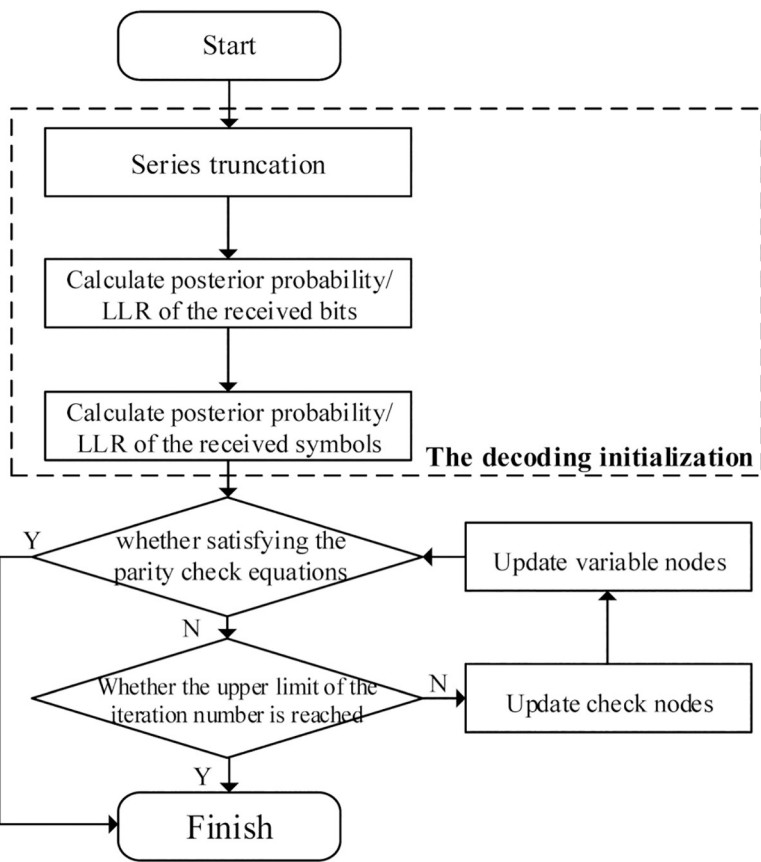

**Fig 1. The flow chart of BP decoding for LDPC codes over GF($q$) on AWAN channel.**

where $P(x_{j,k} = +1) = P(x_{j,k} = -1) = 0.5$. Hence, LLR can be rewritten as

$$
\begin{aligned}
\lambda_{\mathrm{A}}(y_{j,k}) &= \ln \frac{P_{\mathrm{A}}(y_{j,k} - 1)}{P_{\mathrm{A}}(y_{j,k} + 1)} \\
&= \ln \sum_{m=0}^{\infty} \frac{A^m}{m!\sigma_m} \exp(-\frac{(y_{j,k} - 1)^2}{2\sigma_m^2}) - \ln \sum_{m=0}^{\infty} \frac{A^m}{m!\sigma_m} \exp(-\frac{(y_{j,k} + 1)^2}{2\sigma_m^2})
\end{aligned}
\tag{11}
$$

in which $P_{\mathrm{A}}(\cdot)$ denotes the PDF of the Class A noise mentioned in (3). Thus, this PDF is injected in LLR of the incoming BPSK modulated bits.

FFT BP decoding with the proposed initialization (i.e. the optimized FFT BP algorithm) for non-binary LDPC codes on AWAN channel is illustrated in Algorithm 1, where $t$ denotes the $t$-th iteration, $E_n(u)$ is the layered message being $u$ ($u \in$ GF($q$)) and initialized with the channel message $P_n(u)$ in Eq (5), $r_{mn}^{(t)}(u)$ represents the message from a check node (CN$_m$) to a variable node (VN$_n$), $q_{nm}^{(t)}(u)$ indicates the message from VN$_n$ to CN$_m$, and $k_v$ is a normalizing constant to ensure $\sum_x q_{nm}^{(t)}(u) = 1$. $\phi(m)$ is the set of variable nodes connected to CN$_m$, and $\varphi(m)/n$ is the set of variable nodes connected to CN$_m$ except VN$_n$. $\varphi(n)$ is the set of check nodes connected to VN$_n$, and $\varphi(n)/m$ is the set of check nodes connected to VN$_n$ except CN$_m$. $\mathbf{P}(\cdot)$ and $\mathbf{P}^{-1}(\cdot)$ denote the permutation function and inverse permutation function [40]. $\mathbf{F}(\cdot)$ and $\mathbf{F}^{-1}(\cdot)$ indicate the FFT function and IFFT function. $\tilde{q}_{nm}^{(t)}(u)$ denotes the permutation message of $q_{nm}^{(t)}(u)$. $\tilde{Q}_{nm}^{(t)}(u)$ is the Fourier domain message of $\tilde{q}_{nm}^{(t)}(u)$. $\tilde{r}_{mn}^{(t)}(u)$ is the check node

updated message in the probability domain. $\tilde{R}_{mn}^{(t)}(u)$ indicates the check node updated message in the Fourier domain. $r_{mn}^{(t)}(u)$ also indicates the inverse permutation message of $\tilde{r}_{mn}^{(t)}(u)$.    In practice, we need to estimate the impulsive coefficient $A$, GIR $\Gamma$ and the background Gaussian noise power $\sigma_G^2$ properly. Fortunately, these parameters can be estimated from the second, fourth and sixth epochs of the received envelopes [41].

---

**Algorithm 1:** FFT BP decoding with the proposed initialization for non-binary LDPC codes on AWAN channel

---

**Inputs:** an $M{\times}N$ sparse parity-check matrix **H**, the received sequence $y$, iter_max
**Outputs:** $Q$-ary codeword $\hat{u}$

(Initialization)

1: **Evaluate log likelihood ratio (LLR) of the received bits with series truncation.**

$$\lambda_A(y_{j,k}) = \ln\sum_{m=0}^{\infty}\frac{A^m}{m!\sigma_m}\exp(-\frac{(y_{j,k}-1)^2}{2\sigma_m^2}) - \ln\sum_{m=0}^{\infty}\frac{A^m}{m!\sigma_m}\exp(-\frac{(y_{j,k}+1)^2}{2\sigma_m^2})$$

2: **Calculate the posterior probabilities of the received bits by LLR.**

$$P(x_{j,k} = +1 \mid y_{j,k}) = \frac{1}{1+\exp(-1\cdot\lambda_A(y_{j,k}))}, \tag{12}$$

$$P(x_{j,k} = -1 \mid y_{j,k}) = 1 - P(x_{j,k} = 1 \mid y_{j,k}). \tag{13}$$

3: **Compute the posterior probability of the received symbol by multiplying posterior probabilities of several received bits as shown in (5).**

(Iterations)

4: **for all** $t = 1 \rightarrow$ iter_max **do**

5:  **for all** $l = 1 \rightarrow L$ **do**

6:   **for all** $m \in M_l, n \in \varphi(m)$ **do**

7:    Step 1: Update the message $q_{nm}^{(t)}(u) = k_v E_n(u)/r_{mn}^{(t-1)}(u)$

8:    Step 2: Permute the message $\tilde{q}_{nm}^{(t)}(u) = \mathbf{P}(q_{nm}^{(t)}(u))$

9:    Step 3: Transform $\tilde{q}_{nm}^{(t)}(u)$ into the Fourier domain $\tilde{Q}_{nm}^{(t)}(u) = \mathbf{F}(\tilde{q}_{nm}^{(t)}(u))$

10:    Step 4: Update $\tilde{R}_{mn}^{(t)}(u)$ message $\tilde{R}_{mn}^{(t)}(u) = \prod_{n'\in\varphi(m)/n}\tilde{Q}_{n'm}^{(t)}(u)$

11:    Step 5: Transform into the Probability domain $\tilde{r}_{mn}^{(t)}(u) = \mathbf{F}^{-1}(\tilde{R}_{mn}^{(t)}(u))$

12:    Step 6: Apply inverse permutation to $r_{mn}^{(t)}(u) = \mathbf{P}^{-1}(\tilde{r}_{mn}^{(t)}(u))$

13:    Step 7: Update $E_n(u)$ message $E_n(u) = q_{nm}^{(t)}(u)\cdot r_{mn}^{(t)}(u)$

14:   **end for**

15:  **end for**

16:  Perform hard decision to obtain $\hat{u}$ ($\hat{u}_n = argmax_{u\in GF(q)}\{E_n(u)\}$ for $n=1\rightarrow N$)

17:  **if** $\hat{u}$ satisfies the sparse parity-check matrix **H then**

18:   break

19:  **end if**

20: **end for**

**Outputs** $\hat{u}$

---

Furthermore, since the PDF of the Class A noise $P_A(z)$ in (3) consists of infinite series, the calculation of LLR is extremely complex practically. Thus, the series truncation can be regarded as a valid alternative to compute effectively by

$$P_A(z) = \sum_{m=0}^{L} \frac{e^{-A}A^m}{m!} \cdot \frac{1}{\sqrt{2\pi}\sigma_m} \exp\left(-\frac{z^2}{2\sigma_m^2}\right)$$

$$= \sum_{m=0}^{L} f_m(A, \sigma_m, z)$$

(14)

To obtain an appropriate $L$, we need to analyze the PDF of the Class A noise. Given the noise amplitude z, the value of each term in (14) is mainly influenced by impulsive coefficient $A$ and equal to the product of Poisson component and Gaussian component. Fig 2 illustrates the relationship between Poisson component and term number $m$ with different $A$. From Fig 2, we can observe that for the smaller $A$, the term $m$ corresponding to the maximum value of Poisson probability becomes smaller. If $A = 1$, Poisson probability is close to 0 under the condition of $m \geq 4$, which leads to a negligible value for the $m$-th term of $P_A(z)$. Thus, $L$ is considered as the minimal integer whose Poisson cumulative distribution with mean $A$ is more than 99%:

$$\sum_{m=0}^{L} \frac{e^{-A}A^m}{m!} > 99\%.$$

(15)

According to (15), $L$ equals to 3 with $A = 1$. $P_A(z)$ with different $L$ is shown in Fig 3 and we can find that the curves are overlapped with $L>3$, which indicates formula (15) is effective.

After introducing the optimized BP decoding of non-binary LDPC codes on AWAN channel, we investigate the characteristics of LLR on AWGN channel and AWAN channel with $A = 0.1$, $\Gamma = 0.1$. Suppose that the code rate is 1/2 and the signal-to-noise ratio $E_b/N_0$ is 0 dB for the received signal, where $E_b$ denotes the energy per information bit and $N_0$ is the one-sided

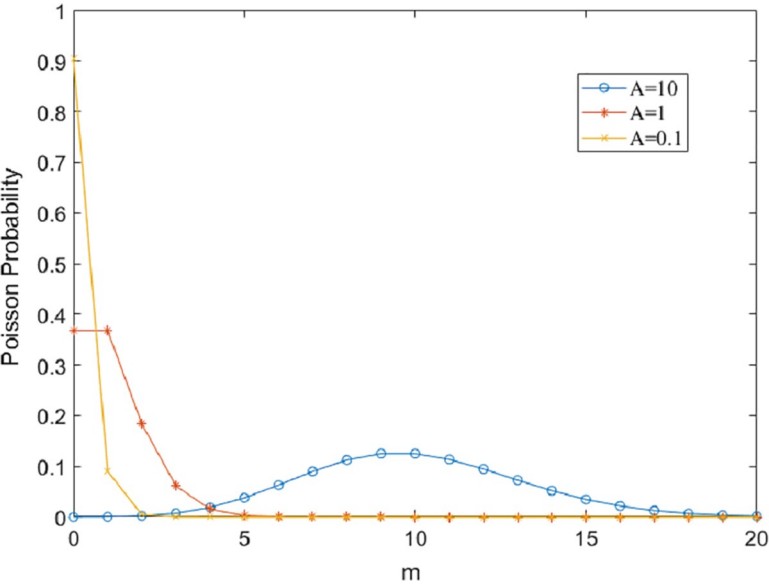

**Fig 2. Poisson probability vs. the term number m with different mean A.**

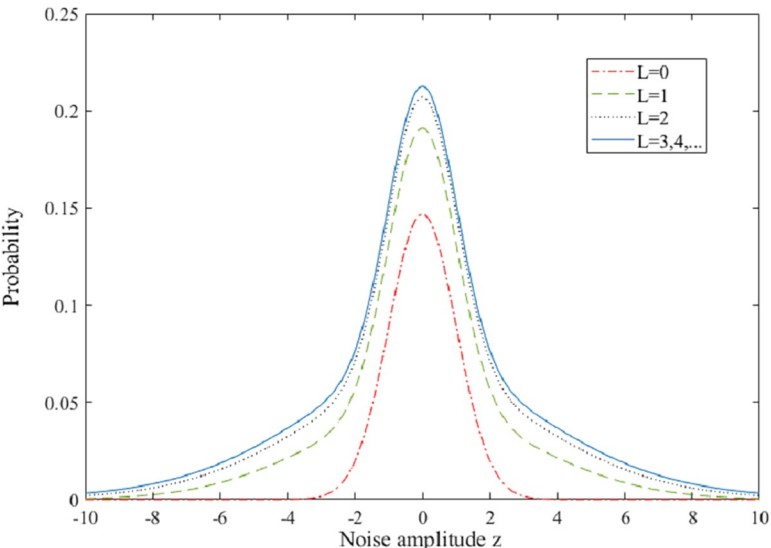

**Fig 3. PDF of noise amplitude z for A = 1 and L = 0, 1, 2, . . ..**

Gaussian noise power spectral density. According to the relationship between $E_b/N_0$ and the background Gaussian noise power $\sigma_G^2$, the following formulas can be acquired:

$$\sigma_G{}^2 = \frac{1}{2R*10^{(0.1E_b/N_0)}} = 1, \tag{16}$$

$$\sigma_I{}^2 = \sigma_G{}^2/\Gamma = 10. \tag{17}$$

Fig 4 depicts the characteristics of LLR with different amplitudes of the received signal on AWGN and AWAN channels. From Fig 4, we can draw the conclusion that, the LLR is

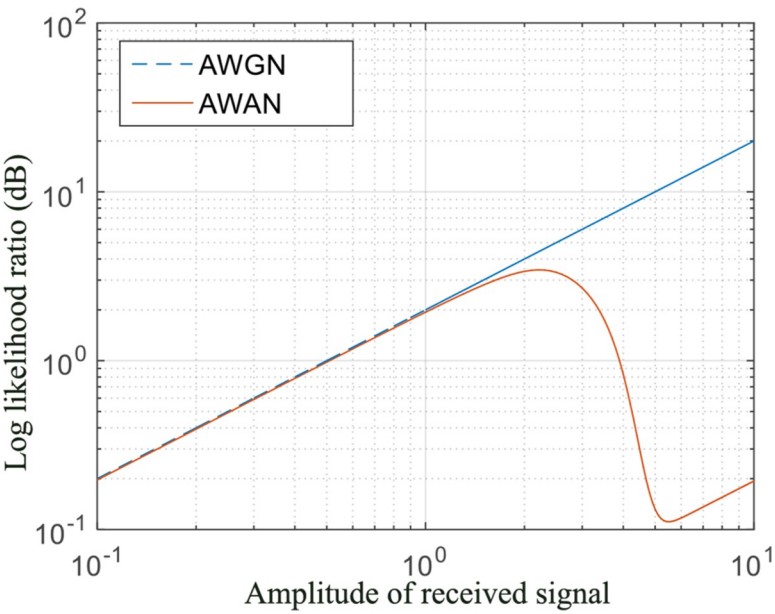

**Fig 4. LLR for AWGN and AWAN channels.**

proportional to the amplitude of the received signal for AWGN channel; while for AWAN channel, it has a nonlinear relationship with the amplitude of the received signal. Specifically, for AWAN channel, the LLR is proportional to the amplitude of the received signal and similar to that of AWGN channel when the signal amplitude is less than 1. On the other hand, as the received signal amplitude increases, the corresponding LLR degrades due to that the received signal may contain impulsive noise with high probability.

## Results and discussion

In this section, series of experiments are conducted to demonstrate the efficient performance of the proposed optimized initialization for LDPC decoding over GF($q$) on AWAN channel. Irregular quasi-cyclic (QC) LDPC codes (88, 44) and (200, 100) over GF(64) with bit rate of 1/2 and a binary irregular QC-LDPC code (528, 264) with bit rate of 1/2 were utilized in our experiments. And the decoder was set to perform at most 50 iterations. Different decoding methods are evaluated and compared with respect to convergence, validity, and robustness. Experimental results indicate the superiority of the proposed optimized initialization.

### Convergence comparison

Firstly, convergence issues have been considered as in [42] and performance of different initializations was evaluated utilizing the BER. To be specific, we compared the BER of the FFT BP decoding algorithm with the optimized initialization (i.e. the optimized FFT BP algorithm) with that of the conventional FFT BP algorithm (designed for AWGN channel) on AWAN channel. We should note that the difference between the two algorithms is the initialization. Therefore, the statistical evaluation was performed using the Wilcoxon rank test metric of both algorithms for QC LDPC codes (88, 44) with the statistical significance value $\alpha = 0.05$. The null hypothesis H0 is 'The difference between $E_b/N_0$ obtained by the optimized FFT BP algorithm and the conventional FFT BP algorithm is identical with the same BER'. Meanwhile, the alternative hypothesis is set as 'the optimized FFT BP algorithm is validated'.

Table 1 presents the convergence comparisons of different initializations using the Wilcoxon Signed-Rank Test metric at different BER, where the '+' indicates the cases when the algorithm acquires better coding gain. It clearly shows that the FFT BP algorithm with the proposed optimized initialization is statistically more superior.

More specifically, as shown in Fig 5, the BER performance of the conventional FFT BP algorithm for the QC LDPC code (88, 44) suffers from large degradation on AWAN channel with $A = 0.1$ and $\Gamma = 0.1$. Actually, in contrast to the BER on AWGN channel, the conventional FFT BP on AWAN channel suffers about 14 dB degradation at BER = $10^{-5}$ because of the emergence of the impulse noise. In particular, assume that the GIR $\Gamma = \sigma_G^2/\sigma_I^2 = 0.1$, then the total noise power $\sigma^2$ of AWAN channel is 11 times of $\sigma_G^2$. If the total noise power on AWAN channel is the same as that on AWGN channel, the performance of the conventional FFT BP on AWAN channel suffers about 3.6 dB degradation than that on AWGN channel at BER = $10^{-5}$.

Further, from Fig 5, we can find that the proposed optimized FFT BP decoding algorithm with series truncation performs efficiently on AWAN channel. More precisely, it achieves about 12.2 dB coding gain at BER = $10^{-5}$ compared to the conventional one.

**Table 1. Results of convergence comparisons at different BER by utilizing Wilcoxon rank test ($\alpha = 0.05$).**

| BER | $10^{-1}$ | $10^{-2}$ | $10^{-3}$ | $10^{-4}$ | $10^{-5}$ |
|---|---|---|---|---|---|
| $p$ value | 0 | 0 | 0 | 0 | 0 |
| The optimized FFT BP | + | + | + | + | + |

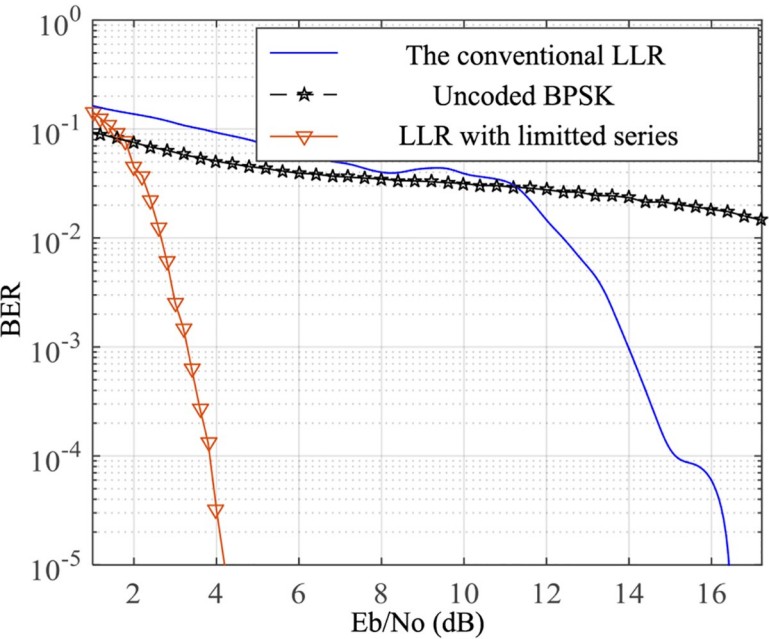

**Fig 5. BER performance of the optimized FFT BP algorithm on AWAN channel.**

## Validity analysis

Secondly, the BER performance of LDPC code (88, 44) over GF(64) was compared with the binary LDPC code (528,264) on AWAN channel with $A = 0.1$, $\Gamma = 0.1$ in Fig 6. The decoding algorithm of binary LDPC codes on AWAN channel was introduced by Nakagawa et al. in [27]. Fig 6 demonstrates that the BER performance of binary LDPC codes on AWAN channel

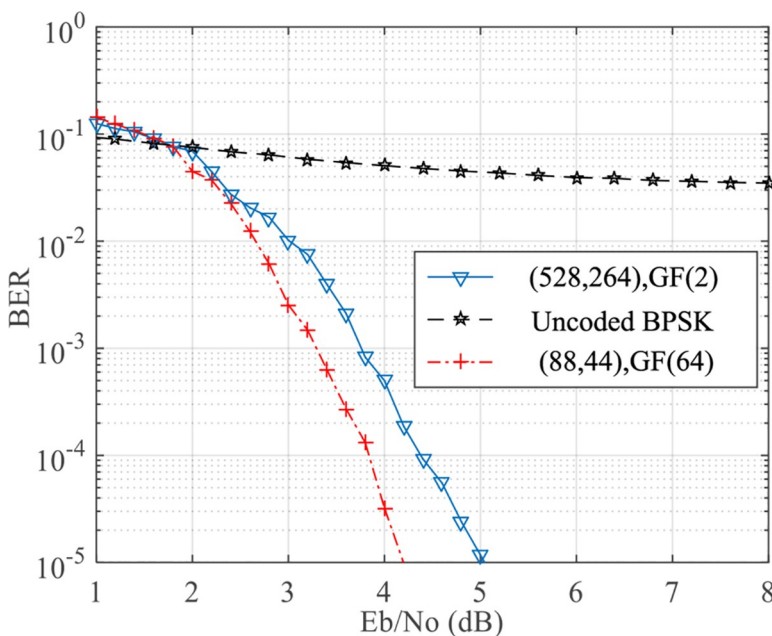

**Fig 6. Performance comparison of LDPC codes over GF(2) and GF(64) on AWAN channel.**

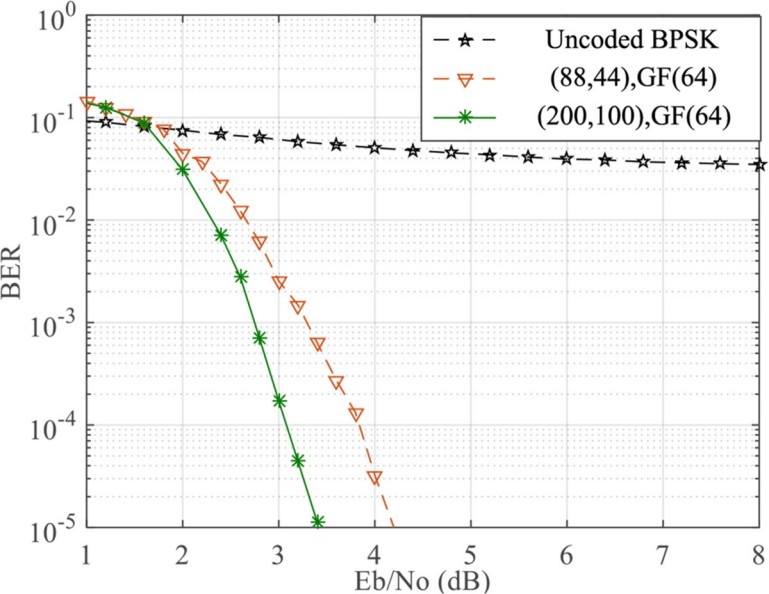

**Fig 7. Performance comparison of QC LDPC codes on AWAN channel with $A$ = 0.1 and $\Gamma$ = 0.1.**

suffers from about 1.7 dB degradation than LDPC codes over GF(64), which reveals that LDPC codes over GF(64) outperform binary LDPC codes on AWAN channel. Although there is an increase in computational complexity for decoding of non-binary LDPC codes, this could be appropriately addressed with the rapid development of hardware in terminals.

## Robustness analysis

Moreover, the robustness of the proposed initialization by considering LDPC codes with different lengths is analyzed. Fig 7 shows the performance comparison of the proposed approach with QC LDPC codes (88, 44) and (200, 100) over GF(64) on AWAN channel with $A$ = 0.1 and $\Gamma$ = 0.1. We can observe that, for both long and short codes, the decoding algorithm with the optimized initialization can be performed accurately and QC LDPC codes (200, 100) has 1 dB coding gain over QC LDPC codes (88, 44) at BER = $10^{-5}$. It can be concluded that, the proposed initialization shows an excellent robustness.

## Effect of channel parameters

We also investigate the effect of different AWAN channel parameters on performance of the proposed FFT BP decoding algorithm. Table 1 summarizes the signal-to-noise ratio $E_b/N_0$ corresponding to BER = $10^{-5}$ under the variation of the impulsive coefficient $A$ and the GIR $\Gamma$ respectively. From this table, it is clear that $E_b/N_0$ becomes larger as the impulsive coefficient $A$ increases. The reason is that, the larger $A$ is, the more continuous impulsive noises would be, i.e., the statistical characteristics of the Class A noise approximate these of the Gaussian noise. Thus, the impulsive noise would not be suppressed well which leads to a worse performance of the optimized decoding algorithm. Particularly, the proposed algorithm with $A$ = 10 suffers from 10.5 dB degradation compared to that with $A$ = 0.01 at BER = $10^{-5}$.

Additionally, Table 2 also shows that $E_b/N_0$ becomes larger as the GIR $\Gamma$ increases from 0.01 to 1. It is because that, with a larger $\Gamma$, the Class A noise shows fewer impulse characteristics and gets similar to the Gaussian noise. Therefore, the impulsive noise would not be

**Table 2. Effect of AWAN channel parameters on performance of the optimized FFT BP.**

| $A$ | 0.01 | 0.1 | 1 | 10 |
|---|---|---|---|---|
| $E_b/N_0$ ($\Gamma = 0.1$) | 2.5 dB | 4.2 dB | 13 dB | 12.9 dB |
| $\Gamma$ | 0.01 | 0.1 | 1 | 10 |
| $E_b/N_0$ ($A = 0.1$) | 3.6 dB | 4.2 dB | 4.35 dB | 2.8 dB |

suppressed well, which results in a worse performance. In contrast, $E_b/N_0$ becomes smaller as the GIR increases to 10. The performance is improved due to the fact that the impulse noise power becomes much small as the GIR increases further. Accordingly, the optimized decoding algorithm with $\Gamma = 10$ performs about 1.4 dB coding gain over that with $\Gamma = 0.1$ at BER = $10^{-5}$.

### Simplification

Series truncation of the PDF of the Class A noise is introduced to simplify the calculation in section 4. Further, we compare it with another approach mentioned in [43] by piecewise fitting the optimal LLR in (11). Consider the LLR is related to the channel parameters, two piecewise functions were chosen to fit the optimal LLR with $\Gamma = 0.1$ and $A = 0.1$, which could be expressed as $F_1 = \sqrt{2} \cdot z/\sigma_G^2$, $F_2 = 70 \cdot \sigma_G^2/z^3$, where $z$ denotes the amplitude of the received signal and $\sigma_G^2$ is the Gaussian noise power of the Class A noise.

From Fig 8, it can be observed that, the BER performance of LLR with series truncation achieves 0.4 dB coding gain over that of LLR with piecewise fitting. Similar to the previous conclusion, the decoding algorithm with the optimized initialization shows superiority over the competing methods. Although the fitting approach makes the calculation easier, unfortunately, it is difficult to derive the relationship between the fitting function and channel parameters. Hence, we have to recalculate the fitting function under the variation of the channel parameters $A$ and $\Gamma$, which complicates the decoding procedure.

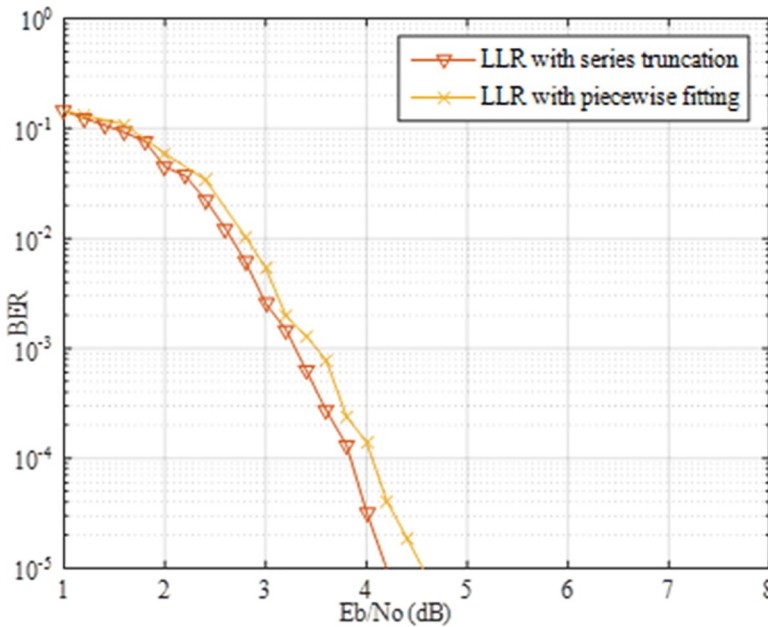

**Fig 8. Performance comparison of the proposed method with piecewise fitting.**

## Conclusion

In this paper, we investigate LDPC decoding over GF($q$) in impulsive noise environments for modern navigation satellite communication. By jointly considering the Class A noise model and the series truncation, we propose an optimized initialization for LDPC decoding over GF ($q$) on AWAN channel, which can be employed in BP-based iterative decoding algorithm. In addition, convergence, validity, and robustness of the proposed initialization are analyzed and discussed with extensive experiments. Simulation results demonstrate that, the decoding algorithm with the optimized initialization achieves 12.2 dB coding gain at BER = $10^{-5}$ compared to conventional methods on the assigned AWAN channel. Moreover, LDPC codes over GF($q$) acquire 1.7 dB coding gain over binary LDPC codes at BER = $10^{-5}$ on AWAN channel. Robustness and the effect of channel parameters are confirmed by considering LDPC codes with different lengths and AWAN channel with parameters. Furthermore, by comparing the proposed method and the piecewise fitting method, experimental result verifies the feasibility of our method in practical applications. With the continues development of BDS-3, to employ the optimized initialization for LDPC decoding over GF($q$) can achieve superior performance significantly in impulsive noise environments. The optimized initialization proposed in this paper can be also extended to decoding process of other 5G or 6G applications. In the future work, we will focus on studying LDPC decoding over GF($q$) in a more complex noise environment for B5G and 6G systems.

## Supporting information

**S1 Data.**
(ZIP)

## Acknowledgments

The authors wish to thank Prof. Qin Huang for the provision of valuable datasets. Thanks also go to the authors of the papers mentioned in REFERENCES.

## Author Contributions

**Methodology:** Haoqiang Liu, Hongbo Zhao, Xiaowen Chen.

**Writing – original draft:** Haoqiang Liu, Hongbo Zhao, Xiaowen Chen.

**Writing – review & editing:** Wenquan Feng.

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
