## [Decision Letter · Decision Letter 0]

13 Jul 2020

PONE-D-20-15556

An optimized initialization for LDPC decoding over GF(q) in impulsive noise environments

PLOS ONE

Dear Dr. Zhao,

Thank you for submitting your manuscript to PLOS ONE. After careful consideration, we feel that it has merit but does not fully meet PLOS ONE’s publication criteria as it currently stands. Therefore, we invite you to submit a revised version of the manuscript that addresses the points raised during the review process.

We look forward to receiving your revised manuscript.

Kind regards,

Dilbag Singh

Academic Editor

PLOS ONE

Journal Requirements:

2. Thank you for including your funding statement; "No"

Reviewers' comments:

Reviewer's Responses to Questions

**Comments to the Author**

1. Is the manuscript technically sound, and do the data support the conclusions?

Reviewer #1: Yes

Reviewer #2: Yes

2. Has the statistical analysis been performed appropriately and rigorously? 

Reviewer #1: Yes

Reviewer #2: No

3. Have the authors made all data underlying the findings in their manuscript fully available?

Reviewer #1: Yes

Reviewer #2: Yes

4. Is the manuscript presented in an intelligible fashion and written in standard English?

Reviewer #1: Yes

Reviewer #2: Yes

5. Review Comments to the Author

Reviewer #1: This is a good contribution, however, when you mention in the abstract: In this paper, an optimized initialization by calculating posterior probabilities of received symbols is proposed for non-binary

LDPC decoding over additive white Class A noise (AWAN) channel... the optimized initialization can be performed by applying different optimization technqiues? do you think that metaheuristics are a good option? can you detail this in the abstract?

In the second paragraph: ...For current navigation system, LDPC codes over GF(q) is adopted in encoding

and decoding. With the development of BeiDou Navigation Satellite System (BDS) 3,

B1C signal and B2a signal have been utilized gradually. The B-CNAV1 navigation

message is transmitted through B1C signal. Each frame of message consists of 3 sub

frames, of which the second sub frame exploits LDPC codes (200,100) over GF(64)

and the third sub frame leverages LDPC codes (88, 44) over GF(64) for encoding [7].

Meanwhile, the B-CNAV2 navigation message is transmitted through B2a signal with

LDPC codes (96, 48) over GF(64). Nowadays, evaluation and analysis of LDPC codes

over GF(q) for navigation satellite communication, especially in harsh environments,

arouse attentions of researchers all over the world....

When you conclude this paragraph, can you detail why in harsh environments,

arouse attentions of researchers all over the world? are there especial circumstances to consider these LDPC and GF the state of the art?

In section: Conventional Initialization in Decoding Process.. you conclude: And the initial messages of BP algorithm over GF(q) could be obtained by bring

(8) into (5)...again, can you detail why BP is better thab GF?, in addition, it seems that GF is the reference to be improved,

In section: An optimized initialization approach for bp decoding

over gf(q) on awan channel, you describe step by step and include simulation results, but can you include a pseudocode? see examples of pseudocode in this paper : Optimizing the kaplan–yorke dimension of chaotic oscillators applying de and pso, 2019

Can you discuss convergence issues on your optimization initialization? see for example a cese of study in this paper: Convergence rates of the efficient global optimization algorithm for improving the design of analog circuits, 2020

Reviewer #2: 1. Results and findings can be improved further. Interesting to improve the conclusion section as well.

2. Add some key references such as: Deep Transfer Learning based Classification Model for COVID-19 Disease, Classification of COVID-19 patients from chest CT images using multi-objective differential evolution--based convolutional neural networks, Efficient Prediction of Drug-drug interaction using Deep Learning Models, Multi-objective particle swarm optimization-based adaptive neuro-fuzzy inference system for benzene monitoring, Improved Particle Swarm Optimization Based Adaptive Neuro-Fuzzy Inference System for Benzene Detection, Fusion of medical images using deep belief networks

3. Add a suitable diagrammatic flow of the proposed model

4. Authors should add atleast 5 references, otherwise it seems out of scope of PLOS one journal.

6. PLOS authors have the option to publish the peer review history of their article (what does this mean?). If published, this will include your full peer review and any attached files.

Reviewer #1: No

Reviewer #2: No

---

## [Author Response · Author response to Decision Letter 0]

7 Aug 2020

We would like to thank the Editor and the reviewers for their valuable comments and feedback that has helped us make improvements to this paper. We give the respond as an attachment "Response_to_Reviewers.doc".

Reviewer#1, Concern # 1: This is a good contribution, however, when you mention in the abstract: In this paper, an optimized initialization by calculating posterior probabilities of received symbols is proposed for non-binary LDPC decoding over additive white Class A noise (AWAN) channel... the optimized initialization can be performed by applying different optimization techniques? Do you think that metaheuristics are a good option? Can you detail this in the abstract?

Author response: Thanks for this comment. In our manuscript, we analyze problems of the existing decoding algorithms and focus on an optimized initialization by calculating posterior probabilities of received symbols or non-binary LDPC decoding on AWAN channel and. The authors agree that we should consider and discuss the metaheuristic as an alternative to make the article more complete. Metaheuristics are widely recognized as efficient approaches for optimization problems. However, as illustrated in [33], the successful application of metaheuristics requires to find a good initial parameter setting, which is a tedious and time consuming task. Moreover, the performance of metaheuristics deteriorates quickly as the dimensionality increases, nevertheless high-dimensional circumstances are extremely common in encoding and decoding. Therefore, in contrast to metaheuristics, the proposed optimized initialization shows superiority in dealing with decoding problems based on BP.

Author action: We have reorganized the manuscript to make it more clearly. We updated by rewriting the manuscript and adding a new section (i.e. Section 2 “Related work”). And we have included the consideration of metaheuristics in details in Section 2 as follows:

This paper is focused on the design of an optimized initialization for non-binary LDPC codes on AWAN channel. In contrast to metaheuristics, our optimized initialization is dedicated to deal with decoding problems based on BP. Generally, as the famous optimization techniques, metaheuristics are widely recognized as efficient approaches for optimization problems, such as particle swarm optimization (PSO) [26, 27] and differential evolution (DE) [28]. Various metaheuristic methods have reported advantages in image segmentation [29], tuning hyper-parameters of deep neural networks [30, 31], and benzene prediction model [32]. However, as illustrated in [33], the successful application of metaheuristics requires to find a good initial parameter setting, which is a tedious and time consuming task. Moreover, the performance of metaheuristics deteriorates quickly as the dimensionality increases, nevertheless high-dimensional circumstances are extremely common in encoding and decoding. Therefore, decoding algorithms based on BP are considered for non-binary LDPC codes on AWAN channel, and an optimized initialization by calculating posterior probabilities of received symbols is proposed.

Reviewer#1, Concern # 2: In the second paragraph: “...For current navigation system, LDPC codes over GF(q) is adopted in encoding and decoding. With the development of BeiDou Navigation Satellite System (BDS) 3, B1C signal and B2a signal have been utilized gradually. The B-CNAV1 navigation message is transmitted through B1C signal. Each frame of message consists of 3 sub frames, of which the second sub frame exploits LDPC codes (200,100) over GF(64) and the third sub frame leverages LDPC codes (88, 44) over GF(64) for encoding [7]. Meanwhile, the B-CNAV2 navigation message is transmitted through B2a signal with LDPC codes (96, 48) over GF(64). Nowadays, evaluation and analysis of LDPC codes over GF(q) for navigation satellite communication, especially in harsh environments, arouse attentions of researchers all over the world....”. When you conclude this paragraph, can you detail why in harsh environments,

arouse attentions of researchers all over the world? Are there especial circumstances to consider these LDPC and GF the state of the art?

Author response: Thanks for this comment. The authors agree that we should illustrate the especial circumstances to consider LDPC and GF the state of the art and describe why navigation communication arouses attentions of researchers in harsh environments.

Author action: We have rewritten the Section 1 in our manuscript and added more descriptions about the above issues in the second and third paragraphs as follows:

Non-binary LDPC codes have been recognized as a powerful technology to encode navigation information efficiently. With the continuous development of BDS 3 and the increasing demand for navigation, people hope to access precise navigation information in time. However, working in severe environments, such as underwater environment surrounded by acoustical noises or in the power industry, the decoder of non-binary LDPC codes is inevitably subject to impulsive interference, which causes poor reliability of transmitted information and even leads to communication failure. Therefore, nowadays, evaluation and analysis of LDPC codes over GF(q) for navigation satellite communication, especially in harsh environments, arouse attentions of researchers all over the world.

Conventionally, the LDPC decoding algorithms are designed for AWGN channel. However, navigation satellite communication often suffers from irregular noises due to the occurrence of high amplitude “spikes”. For navigation satellite communication, such spikes can be generated in atmosphere where lightning discharges in the vicinity of the receiver, or underwater environment where the ambient acoustical noises includes impulses due to noisy aquatic animals such as snapping shrimp [8-11]. At this time, the original assumption that noises are the Gaussian noises is inadequate. Unfortunately, few attention has been given to decoding algorithms for LDPC codes over GF(q) in impulsive noise environments.

Reviewer#1, Concern # 3: In section: Conventional Initialization in Decoding Process. You conclude: And the initial messages of BP algorithm over GF(q) could be obtained by bring (8) into (5) ...again, can you detail why BP is better than GF? In addition, it seems that GF is the reference to be improved,

Author response: Thanks for this comment. As illustrated in our manuscript, a significant amount of research has been concentrated on the design, encoding, decoding and performance analysis of LDPC codes over GF(q). And iterative decoding algorithm based on BP is an important soft decision decoding algorithm. In section “Conventional Initialization in Decoding Process”, we give the conventional initialization of BP decoding algorithm considering non-binary LDPC codes. Non-binary LDPC codes mean LDPC codes over GF(q). By adding (8) to (5), the initial messages of BP algorithm for non-binary LDPC codes could be obtained. 

Author action: We have enhanced the detailed description about why non-binary LDPC codes perform better compared to binary LDPC codes in Section 1 (the penultimate paragraph) and Section 3 as follows:

Generally, compared with binary LDPC, LDPC codes over GF(q) show superior performance in resisting burst errors such as impulsive noises for the characteristic of inner interleaving [12, 13], which makes non-binary LDPC codes more suitable for navigation communication. Furthermore, non-binary LDPC codes combined with q-ary modulation can increase transmission rate obviously [14, 15]. Although there is an increment of computational complexity by adopting LDPC codes over GF(q), with the development of terminal computation, non-binary LDPC codes have become a hot research topic and would become more prevalent and applicable. Therefore, we focus on non-binary LDPC codes decoding on additive white Class A noise (AWAN) channel and present an optimized initialization for decoding in this paper. With simulation experiments, we demonstrate the efficiency of the proposed algorithm.

Non-binary LDPC codes can be considered as a kind of linear block codes which are the extension of binary LDPC codes over GF(q). The difference between non-binary LDPC codes and binary LDPC codes is that each non-zero element of sparse parity-check matrix needs to be obtained from GF(q). The Tanner graph of non-binary LDPC codes given by a sparse parity-check matrix over GF(q) is constructed in the same way as that of binary LDPC codes. Compared with binary LDPC codes, non-binary LDPC codes perform better in communication due to advantages in resisting burst error and high transmission rate [34-37].

The GF(q) related references include [12-15, 34-37].

Also, we have enhanced the detailed description about BP decoding and initialization in Section 3 as:

The conventional decoding methods degrade severely while facing the impulsive noise since those methods acquire the posterior probability of received symbols by making use of the transition probability on AWGN channel. To tackle this problem, based on the BP decoding algorithm, we present an optimized initialization for LDPC decoding over GF(q) on AWAN channel with series truncation.

As mentioned above, the initialization is the crucial part of the BP decoding over GF(q) algorithm. The posterior probability of received symbols is considered as the initial message transmitted from the variable nodes to the check nodes. Thus, to evaluate the posterior probability of received symbols correctly is vital for decoding.

During the initialization of conventional decoding process, we obtain the posterior probability of the received bits from the transition probability of the AWGN channel as equation (9), which suffers from degradation of the bit error rate (BER) in impulsive noise environments. Therefore, for LDPC decoding over GF(q) on AWAN channel, since the channel parameters are only leveraged in the decoding initialization, we need to improve the initialization of the iterative decoding process without changing the information transmission mode between variable nodes and check nodes. Under this circumstance, the optimized initialization can be applied to various traditional LDPC decoding algorithms over GF(q), such as FFT BP and EMS algorithm. Generally, the BP decoding algorithm with optimized initialization for LDPC codes over GF(q) on AWAN channel can be illustrated by the flow chart in Fig 1.

In addition, Figure 1 has been redrawn as follows and new references related to BP (such as Carrasco R A, Johnston M. Non-binary error control coding for wireless communication and data storage. John Wiley & Sons) have been added:

Reviewer#1, Concern # 4: In section: An optimized initialization approach for BP decoding

over GF(q) on AWAN channel, you describe step by step and include simulation results, but can you include a pseudocode? See examples of pseudocode in this paper: Optimizing the kaplan–yorke dimension of chaotic oscillators applying de and pso, 2019

Author response: Thanks for this comment very much. The authors agree that the pseudocode of the proposed initialization is significantly important.

Author action: The authors have read the paper “Optimizing the kaplan–yorke dimension of chaotic oscillators applying de and pso” carefully. The pseudocode in this paper seems very clear and powerful. Moreover, this paper introduces the optimization of the Kaplan-Yorke dimension of chaotic oscillators by applying metaheuristics such as DE and PSO algorithms, which is consistent with our discussion about metaheuristics in Section 2. 

We updated the manuscript by adding this paper as reference. And we have included the symbol description and pseudocode of FFT BP decoding with the proposed initialization for non-binary LDPC codes on AWAN channel instead of the original descriptions step by step in Section 4 as:

FFT BP decoding with the proposed initialization (i.e. the optimized FFT BP algorithm) for non-binary LDPC codes on AWAN channel is illustrated in Algorithm 1, …..(in the 5-th paragraph in Section 4)

And the correspond pseudocode is added as:

Reviewer#1, Concern # 5: Can you discuss convergence issues on your optimization initialization? See for example a case of study in this paper: Convergence rates of the efficient global optimization algorithm for improving the design of analog circuits, 2020.

Author response: Thanks for this comment very much. The authors agree that there should be more discussions with respect to the convergence of the proposed algorithm. We have conducted researches on the recommended article “Convergence rates of the efficient global optimization algorithm for improving the design of analog circuits” and found that convergence rates and comparisons performed by this article were useful.

Author action: We have conducted researches on the recommended articles and performed experiments with respect to the convergence. We have reorganized the manuscript and added a new subsection “Convergence comparison” in Section 5 “Results and discussion” as follows:

Firstly, convergence issues have been considered as in [40] and performance of different initializations was evaluated utilizing the BER. To be specific, we compared the BER of the FFT BP decoding algorithm with the optimized initialization (i.e. the optimized FFT BP algorithm) with that of the conventional FFT BP algorithm (designed for AWGN channel) on AWAN channel. We should note that the difference between the two algorithms is the initialization. Therefore, the statistical evaluation was performed using the Wilcoxon rank test metric runs of both algorithms for QC LDPC codes (88, 44) with the statistical significance value. The null hypothesis H0 is ‘The difference between obtained by the optimized FFT BP algorithm and the conventional FFT BP algorithm is identical with the same BER’. Meanwhile, the alternative hypothesis is set as ‘the optimized FFT BP algorithm is validated’. 

Table 1 presents the convergence comparisons of different initializations using the Wilcoxon Signed-Rank Test metric at different BER, where the ‘+’ indicates the cases when the algorithm acquires better coding gain. It clearly shows that the FFT BP algorithm with the proposed optimized initialization is statistically more superior.

And results of convergence comparisons at different BER by utilizing Wilcoxon rank test is also added as Table 1.

Reviewer#2, Concern # 1: Results and findings can be improved further. Interesting to improve the conclusion section as well.

Author response: Thanks for this comment very much. The authors agree that both section should be improved further. We have rewritten Section 5 “Results and discussion” and Section 6 “Conclusion” 

Author action: We have updated by rewriting all experimental results to make them clear and adding a new subsection (i.e. convergence comparison) in Section 5, which has been described in Reviewer#1, Concern # 5.

Further, we have rewritten Section 6 “Conclusion” as:

In this paper, we investigate LDPC decoding over GF(q) in impulsive noise environments for modern navigation satellite communication. By jointly considering the Class A noise model and the series truncation, we propose an optimized initialization for LDPC decoding over GF(q) on AWAN channel, which can be employed in BP-based iterative decoding algorithm. In addition, convergence, validity, robustness of the proposed initialization are analyzed and discussed with extensive experiments. Simulation results demonstrate that, the decoding algorithm with the optimized initialization achieves 12.2 dB coding gain at BER=10-5 compared to conventional methods on the assigned AWAN channel. Moreover, LDPC codes over GF(q) acquire 1.7 dB coding gain over binary LDPC codes at BER = 10-5 on AWAN channel. Robustness and the effect of channel parameters are confirmed by considering LDPC codes with different lengths and AWAN channel with parameters. Furthermore, by comparing the proposed method and the piecewise fitting method, experimental result verifies the feasibility of our method in practical applications. With the continues development of BDS-3, to employ the optimized initialization for LDPC decoding over GF(q) can achieve superior performance significantly in impulsive noise environments. In the future work, we will focus on studying LDPC decoding over GF(q) in a more complex noise environment.

Reviewer#2, Concern # 2: Add some key references such as: Deep Transfer Learning based Classification Model for COVID-19 Disease, Classification of COVID-19 patients from chest CT images using multi-objective differential evolution--based convolutional neural networks, Efficient Prediction of Drug-drug interaction using Deep Learning Models, Multi-objective particle swarm optimization-based adaptive neuro-fuzzy inference system for benzene monitoring, Improved Particle Swarm Optimization Based Adaptive Neuro-Fuzzy Inference System for Benzene Detection, Fusion of medical images using deep belief networks

Author response: Thanks for this comment very much. The authors agree that there should be more related references in this manuscript. Since our manuscript introduces an optimized initialization, as the famous optimization techniques, the authors agree that the metaheuristic should be considered and discussed as an alternative to make the article more complete. We have conducted researches on the recommended articles and found that those excellent articles are closely related to our consideration of metaheuristic algorithms. “Classification of COVID-19 patients from chest CT images using multi-objective differential evolution--based convolutional neural networks” presents a CNN to classify the chest CT images as as infected (+ve) or not (-ve) and tunes the initial parameters of CNN with multi-objective differential evolution. “Deep Transfer Learning based Classification Model for COVID-19 Disease” and “Fusion of medical images using deep belief networks” combine deep learning with meta-heuristics and give a detailed description about tuning hyper-parameters of deep neural networks. “Improved Particle Swarm Optimization Based Adaptive Neuro-Fuzzy Inference System for Benzene Detection” extends the metaheuristic algorithm to benzene prediction model. “Multi-objective particle swarm optimization-based adaptive neuro-fuzzy inference system for benzene monitoring” employs the metaheuristic (PSO) to enhance the accuracy of ANFIS for runtime parameter.

Author action: We have conducted researches on the recommended articles and included those excellent literature as our references. We updated by rewriting the manuscript and adding a new section (i.e. Section 2 “Related work”). We have included the consideration of metaheuristics in details with the recommended articles as important references in Section 2 as:

Generally, receivers adopts parameters of the AWGN channel and conventional decoding methods for LDPC decoding, which results in serious degradation in impulsive noise environments. The impulsive component of interference has been found to be significant which influences the reliability of transmitted information.

Various attempts have been made to develop models of impulsive noises that can be divided into empirical models and physical models. Class A noise model proposed by Middleton is a typical kind of physical models [23]. The statistical feature of Class A noise is much different from that of Gaussian noise, therefore the LDPC decoding algorithms for AWGN channel are not suitable for Class A noise environments. Maad et al. analyzed the performance of LDPC codes in heavy-tailed, symmetric alpha stable noise (SαS) channels [24]. Further, Nakagawa et al. proposed the sum-product decoding method for binary LDPC codes in Class A noise environment [25]. However, few researches have ever explored decoding algorithms for LDPC codes over GF(q) in Class A noise environments.

This paper is focused on the design of an optimized initialization for non-binary LDPC codes on AWAN channel. In contrast to metaheuristics, our optimized initialization is dedicated to deal with decoding problems based on BP. Generally, as the famous optimization techniques, metaheuristics are widely recognized as efficient approaches for optimization problems, such as particle swarm optimization (PSO) [26, 27] and differential evolution (DE) [28]. Various metaheuristic methods have reported advantages in image segmentation [29], tuning hyper-parameters of deep neural networks [30, 31], and benzene prediction model [32]. However, as illustrated in [33], the successful application of metaheuristics requires to find a good initial parameter setting, which is a tedious and time consuming task. Moreover, the performance of metaheuristics deteriorates quickly as the dimensionality increases, nevertheless high-dimensional circumstances are extremely common in encoding and decoding. Therefore, decoding algorithms based on BP are considered for non-binary LDPC codes on AWAN channel, and an optimized initialization by calculating posterior probabilities of received symbols is proposed.

Reviewer#2, Concern # 3: Add a suitable diagrammatic flow of the proposed model

Author response: Thanks for this comment. The authors agree that the pseudocode of the proposed initialization is significantly important. 

Author action: We redrawn Fig. 1 (the flow chart of BP decoding for LDPC codes over GF(q) on AWAN channel) and added the pseudocode of the proposed model in our manuscript, which has been described in Reviewer#1, Concern # 4.

Also, to make the manuscript more clear and complete, we have enhanced Section 1 with a detailed description of the contribution as follows:

Therefore, we focus on non-binary LDPC codes decoding on additive white Class A noise (AWAN) channel and present an optimized initialization for decoding in this paper. With simulation experiments, we demonstrate the efficiency of the proposed algorithm. The main contributions are summarized as follows:

1) We investigate the problem of LDPC decoding in impulsive noise environments for navigation communication and formalize the impulsive noise as the Class A noise model.

2) We propose an optimized initialization by calculating posterior probabilities of received symbols for non-binary LDPC decoding on AWAN channel, which makes use of series truncation for computing effectively.

3) Extensive experiments are conducted on convergence, validity, robustness. The experimental results reveal that the optimized initialization has a significant effect on the decoding performance for non-binary LDPC codes on AWAN channel.

Reviewer#2, Concern # 4: Authors should add at least 5 references, otherwise it seems out of scope of PLOS one journal.

Author response: Thanks for this comment. The authors agree that there should be more related references in this manuscript. We have conducted researches on the recommended articles.

Author action: We have conducted researches on the recommended articles and included those excellent literature as our references, which has been described in Reviewer#2, Concern # 2.

---

## [Decision Letter · Decision Letter 1]

8 Apr 2021

PONE-D-20-15556R1

An optimized initialization for LDPC decoding over GF(q) in impulsive noise environments

PLOS ONE

Dear Dr. Zhao,

Thank you for submitting your manuscript to PLOS ONE. After careful consideration, we feel that it has merit but does not fully meet PLOS ONE’s publication criteria as it currently stands. Therefore, we invite you to submit a revised version of the manuscript that addresses the points raised during the review process.

We look forward to receiving your revised manuscript.

Kind regards,

Lisu Yu, Ph.D

Nanchang University, China

Academic Editor

PLOS ONE

Journal Requirements:

Additional Editor Comments (if provided):

The paper seems to be revised well. All the comments are addressed. There is one minor revisions to be noticed. LDPC decoding is a very old technique, so please add some new backgroud and references for that, like 5G or 6G application related references. For example,

[1] “Massively Distributed Antenna Systems With Nonideal Optical Fiber Fronthauls: A Promising Technology for 6G Wireless Communication Systems,” IEEE Vehicular Technology Magazine, Dec. 2020. 

[2] "What should 6G be?" Nature Electronics, Jan. 2020.

and so.

Reviewers' comments:

Reviewer's Responses to Questions

**Comments to the Author**

1. If the authors have adequately addressed your comments raised in a previous round of review and you feel that this manuscript is now acceptable for publication, you may indicate that here to bypass the “Comments to the Author” section, enter your conflict of interest statement in the “Confidential to Editor” section, and submit your "Accept" recommendation.

Reviewer #1: All comments have been addressed

2. Is the manuscript technically sound, and do the data support the conclusions?

Reviewer #1: Yes

3. Has the statistical analysis been performed appropriately and rigorously? 

Reviewer #1: Yes

4. Have the authors made all data underlying the findings in their manuscript fully available?

Reviewer #1: Yes

5. Is the manuscript presented in an intelligible fashion and written in standard English?

Reviewer #1: Yes

6. Review Comments to the Author

Reviewer #1: the revised paper can be accepted now, the authors have addressed all recommendations of this reviewer

7. PLOS authors have the option to publish the peer review history of their article (what does this mean?). If published, this will include your full peer review and any attached files.

Reviewer #1: No

---

## [Author Response · Author response to Decision Letter 1]

14 Apr 2021

Thanks editors. The experimental data involved in this manuscript have been uploaded as the Supporting Information files regarding to each experiment (saved in MATLAB data format, i.e. "*.mat").

Additional Editor Comments (if provided): The paper seems to be revised well. All the comments are addressed. There is one minor revisions to be noticed. LDPC decoding is a very old technique, so please add some new background and references for that, like 5G or 6G application related references. For example, [1] "Massively Distributed Antenna Systems with Nonideal Optical Fiber Fronthauls: A Promising Technology for 6G Wireless Communication Systems," IEEE Vehicular Technology Magazine, Dec. 2020, [2] "What should 6G be?" Nature Electronics, Jan. 2020; and so.

Author response: Thanks for this comment very much. The authors agree that there should be more background and references related to 5G or 6G application in our manuscript. As LDPC decoding is an old technique, we hope to highlight the connection between this paper and the latest communication system.

Author action: We have made the manuscript more clearly. We updated by including some new background in Section 1 “Introduction” and adding some related references to emphasize the 5G or 6G application as follows:

With the explosive development of communication technology, the new mobile communication systems, such as beyond fifth generation (B5G) and sixth generation (6G) systems, will suffer from severe challenges imposed by the requirement for heavy connection density and high efficiency [1]. Especially for 6G, satellite communications play an important role in providing high quality communication services to achieve the worldwide connectivity [2]. As one of the critical components, …

[1]. Yu L, Wu J, Zhou A, et al. Massively Distributed Antenna Systems With Nonideal Optical Fiber Fronthauls: A Promising Technology for 6G Wireless Communication Systems. IEEE Vehicular Technology Magazine. 2020; 15(4): 43-51.

[2]. Dang S, Amin O, Shihada B, et al. What should 6G be?. Nature Electronics. 2020; 3(1): 20-29.

Further, in Section 6 “Conclusion”, we have included the consideration of the next communication systems as follows:

The optimized initialization proposed in this paper can be also extended to decoding process of other 5G or 6G applications. In the future work, we will focus on studying LDPC decoding over GF(q) in a more complex noise environment for B5G and 6G systems

---

## [Editor Report · Decision Letter 2]

19 Apr 2021

An optimized initialization for LDPC decoding over GF(q) in impulsive noise environments

PONE-D-20-15556R2

Dear Dr. Zhao,

We’re pleased to inform you that your manuscript has been judged scientifically suitable for publication and will be formally accepted for publication once it meets all outstanding technical requirements.

Kind regards,

Lisu Yu, Ph.D

Nanchang University

Academic Editor

PLOS ONE

Additional Editor Comments (optional):

No more comments.

Reviewers' comments:

No more comments.

---

## [Editor Report · Acceptance letter]

22 Apr 2021

PONE-D-20-15556R2 

An optimized initialization for LDPC decoding over GF(q) in impulsive noise environments 

Dear Dr. Zhao:

I'm pleased to inform you that your manuscript has been deemed suitable for publication in PLOS ONE. Congratulations! Your manuscript is now with our production department. 

Kind regards, 

on behalf of

Prof. Dr. Lisu Yu 

Academic Editor

PLOS ONE